# Evaluation of the Elements of Short Hairpin RNAs in Developing shRNA-Containing CAR T Cells

**DOI:** 10.3390/cancers15102848

**Published:** 2023-05-20

**Authors:** Ryan Urak, Brenna Gittins, Citradewi Soemardy, Nicole Grepo, Lior Goldberg, Madeleine Maker, Galina Shevchenko, Alicia Davis, Shirley Li, Tristan Scott, Kevin V. Morris, Stephen J. Forman, Xiuli Wang

**Affiliations:** 1Department of Hematology and Hematopoietic Cell Transplantation, City of Hope National Medical Center, Duarte, CA 91010, USA; 2Center for Gene Therapy, Beckman Research Institute, Duarte, CA 91010, USA; 3Menzies Health Institute Queensland, School of Pharmacy and Medical Science, Griffith University, Gold Coast Campus, Southport, QLD 4215, Australia

**Keywords:** chimeric antigen receptor (CAR), short hairpin RNA (shRNA), hypoxanthine phosphoribosyltransferase 1 (HPRT), C-C chemokine receptor type 5 (CCR5), Tat, Rev, shRNA-containing CARs, Patients Living With HIV (PLWH)

## Abstract

**Simple Summary:**

Knocking down genes by shRNAs in CAR T cells offers the potential of expanding the therapy’s efficacy beyond its initial success. Since shRNAs are in the CAR construct, only the CAR specifically will be affected by the knockdown. Due to that intrinsic nature, we show that these knockdowns can make CAR T cells resistant to HIV and chemo-agents. Like other gene editing tools, such as CRISPR, shRNAs need to be optimized. In this article, we elucidate four common design optimizations necessary to construct a fully functional shRNA-containing CAR.

**Abstract:**

Short hairpin RNAs (shRNAs) have emerged as a powerful tool for gene knockdown in various cellular systems, including chimeric antigen receptor (CAR) T cells. However, the elements of shRNAs that are crucial for their efficacy in developing shRNA-containing CAR T cells remain unclear. In this study, we evaluated the impact of different shRNA elements, including promoter strength, orientation, multiple shRNAs, self-targeting, and sense and antisense sequence composition on the knockdown efficiency of the target gene in CAR T cells. Our findings highlight the importance of considering multiple shRNAs and their orientation to achieve effective knockdown. Moreover, we demonstrate that using a strong promoter and avoiding self-targeting can enhance CAR T cell functionality. These results provide a framework for the rational design of CAR T cells with shRNA-mediated knockdown capabilities, which could improve the therapeutic efficacy of CAR T cell-based immunotherapy.

## 1. Introduction

Chimeric antigen receptor (CAR) therapy has demonstrated remarkable success in treating hematological malignancies, such as acute lymphoblastic leukemia (ALL), lymphoma, and multiple myeloma (MM) [1,2,3,4]. However, expanding CAR therapy to non-hematological diseases has proven challenging due to harsh tumor microenvironments, exhaustion, viral infections, and fratricide [1,5,6]. To overcome these obstacles, the field has turned to the genetic modification of CAR T cells.

While the CAR itself is used to genetically alter T cells, the field has also utilized CRISPR technology to suppress genes that may impede CAR functionality [7,8]. The most common genes are the exhaustion molecules, such as PD-1, or tumor microenvironment-associated receptors, such as TGF-β receptor II [9,10]. CRISPR is a nuclease that when combined with a guide RNA can cut a target strand of the T cell DNA leading to insertion or deletions, which may cause the gene to become inactive. While revolutionary to the field, CRISPR and guide RNA are generally transfected into the cells, which leads to two potential problems with CAR T cells. Firstly, the most common strategy of introducing the CRISPR Cas9 protein and guide RNA into the T cells is via electroporation and while electroporation efficiently introduces the editing system into the cells, its impact on the T cells’ viability and growth is not well-established [11,12,13]. Secondly, although CRISPR can make multiple edits to the CAR T cells, the editing efficiency dramatically decreases, which can limit its potential for assisting CAR T cells with all the harsh conditions the cells may come into contact with [14]. With this in mind, we were intrigued to see if there is a potential alternative to CRISPR.

Short hairpin RNAs (shRNAs) have the capacity to suppress gene function by targeting and degrading the mRNA transcripts, preventing the expression of the protein [15]. Unlike CRISPR, shRNAs, single or multiple can be incorporated into the CAR construct/lentivirus, allowing all transduced cells to be knocked down [16,17]. Although shRNAs provide a clear benefit to CAR T cells, shRNAs are not without their faults. Many have reported that including shRNAs within the CAR construct can decrease the transduction efficiency of the CAR lentivirus, which is most likely due to the shRNA having self-targeting leading to decreased lentivirus production [18,19,20]. Moreover, although shRNAs have been introduced into CAR constructs before, there is no consensus on how to design shRNA-containing CAR T cells [17,18,19,21,22]. In this study, we interrogate common design elements of incorporating shRNAs into CAR constructs, such as promoter strength, orientation, and multiple shRNAs, to determine which elements may negatively affect CAR T cells. Additionally, we offer a potential solution to shRNA’s self-targeting without impacting the function of the shRNA. 

## 2. Materials and Methods

### 2.1. DNA Constructs

The CD19 and N6 second-generation (CD19R:CD28:ζ/huEGFRt and N6R:41bb: ζ/huEGFRt) CAR constructs were generated as previously described [4,23]. The addition of CD19 truncation (CD19t) was generated by replacing huEGFRt with gBlock (IDT) of the CD19t using HiFi assembly insertion into the RSRII and NotI site of the second-generation plasmid. The CCR5, CCR5w, HPRT, and Tat/Rev shRNAs were generated by inserting gBlock (IDT) (Table 1) into the NruI and AsiSI sites of the second-generation CAR construct using the Gibson assembly master (NEB).

Infectious molecular clones, pLAI, pBal and pNL4-3, were obtained from NIH AIDS reagent program [24], while pYu2 was obtained from Dr. Jonsson at City of Hope.

### 2.2. Staining and Flow Cytometry 

For non-HIV infected samples, cells were resuspended in phosphate buffer saline (PBS, FisherScientific, Hampton, VA, USA) containing 2% FBS. Cells were washed twice prior to incubation with antibodies for 15 min in the dark at 4 °C. Fluorochrome-conjugated monoclonal antibodies are CD3 (BD Bioscience, San Jose, CA, USA, 563109, 557832), CD4 (BD Bioscience, 557852, 340133), CD8 (BD Biosciences, 348793), CD62L (BD Biosciences, 341012), CD127 (Biolegend, San Diego, CA, USA, 351319), EGFR (Biolegend, 352906), LAG3 (LSBio, Seattle, DC, USA, LS-B2237), PD1 (Invitrogen, Waltham, MA, USA, 47-2799-42), TIM3 (R&D Systems, FAB2365P), CD45RA (BD Biosciences, 555488), CD45RO (BD Biosciences, 561137), CD45 (BD Biosciences, 340665), CD107a (BD Biosciences, 555800), CD45 (BD Biosciences, 340665), or CD19 (Life Technologies, Carlsbad, CA, USA, MHCD1905). Cells were washed twice and immediately before analyzing 1/3 volume of 4’,6-diamidino-2-phenylindole (DAPI), which was added to each sample for viability.

For intracellular staining, cells were resuspended in PBS and washed twice prior to incubation with FVD Viability Viogreen (Thermo Fisher Scientific, Irwindale, CA, USA) at 1:1000 dilution for 15 min at 4 °C. The cells were washed, fixed, and permeabilized with Cytofix/cytoperm plus (BD Bioscience, San Jose, CA, USA) for 20 min at 4 °C. Cells were then stained with intracellular antibodies against interferon gamma (BD Bioscience) and TNF alpha (BD Bioscience San Jose USA) for 15 min at 4 °C.

For HIV infected samples, cells were resuspended in PBS. Cells were washed twice prior to incubation with Zombie Fixable Viability Dye (Biolegend, San Diego, CA, USA) for 20 min at room temperature. Cells were then washed twice with PBS containing 2% FBS and incubated with antibodies for 30 min in the dark at 4 °C. Cells were washed twice and incubated with 1% formaldehyde for 20 min at 4 °C.

### 2.3. Cell Lines

Jurkats (ATCC TIB-152), LCL (ATCC CRL-1805), KG-1a (ATCC CCL-246.1), CEM (ATCC CRL-2265), and 8e5 (ATCC CRL-8993) cell lines were obtained from ATCC^®^. Cells were cultured in RPMI 1640 supplemented with 10% heat inactivated fetal bovine serum (FBS) (Gemini). SupB15 (ATCC CRL-1929) and KG-1a (ATCC CCL-246.1) were cultured in IMDM supplemented with 10% heat inactivated fetal bovine serum (FBS). HEK293 (ATCC CRL-11268) were obtained from ATCC^®^, while HEK293 cells expressing gp160 (HEK.gp160) from the HIV strain 92UG037.8 were provided by Dr. Chen [25]. Cells were maintained in DMEM supplemented with 10% heat inactivated FBS. Cell lines were transduced to express GFP and Firefly luciferase (ffluc) with lentiviral vector encoding for epHIV-7-eGFP-ffluc, and then they were sorted using the BD FACSAria™ SORP cell sorter (BD Bioscience) for >98% purity [4]. LCLs with OKT3-2A-hygromycin_pEK was generated as previously described [26].

### 2.4. Lentiviral and HIV Production

The CAR lentivirus was generated as previously described [4]. HEK293 cells were seeded at 20 × 10^6^ cells/T225 flask for 8 h prior to transfection. Cells were co-transfected with CAR plasmid (47.45 µg), GagPol plasmid (40 µg), Rev plasmid (4 µg), and VSV g plasmid (8 µg) with 2.5M calcium phosphate and 2 × Hank’s balance salt solution [27]. Following a 12-h incubation, media was replaced with DMEM with 2% FBS and 1:1000 sodium butyrate. Following 72 h incubation, supernatant was collected, centrifuged at 2000 rpm for 20 min, filtered through a 0.45-micron filter to remove cell debris, and precipitated by adding 40% (w/v) PEG solution for 12 h. The precipitated virus was pelleted by centrifuging at 2000 × g for 20 min, resuspending the pellet in PBS, and ultra-centrifuging at 100,000× *g* for 1.5 h. Concentrated virus was resuspended in PBS and frozen at −80 °C for later use. CAR lentivirus titers, as determined by EGFRt or CD19t, were quantified by transducing 0.1 × 10^6^ Jurkats cells with various dilutions of virus (10, 30, and 90 µL of 1:200 diluted virus and 1 µL crude virus) and incubated for 7 days prior to flow cytometric analysis.

X4-tropic HIV-1 subtype B (LAI, Yu2, and NL4-3) or R5-tropic HIV-1 (Bal) were generated by transfecting infectious molecular clone (20 µg) with lipofectamine 2000^TM^ (Life Technologies, Carlsbad, CA, USA) into HEK293 cells (4 × 10^6^ cells/T175). Cells were incubated for 48–72 h before supernatant was collected, centrifuged at 1500× *g*, filtered through 0.45-micron filter, and frozen at −80 °C until later use as described previously [28,29]. The viral titer was calculated using a TZM-bl assay [30].

### 2.5. Generation of CAR T Cells

Whole blood from healthy donors was obtained from the City of Hope blood donor center and HIV whole blood was obtained from ZenBio. Peripheral blood mononuclear cells (PBMCs) were isolated using Ficoll-Paque (Amersham Biosciences, Slough, United Kingdom) density gradient centrifugation [27] and enriched for CD3+ cells by EasySep™ Human T cell Isolation Kit (StemCell Technologies, Vancover, BC, Canada). Enriched CD3+ T cells were activated with dynabeads™ human T-activator CD3/CD28^®^ (Life Technologies Carlsbad USA) and transduced with CAR lentivirus at MOI 2. CAR T cells were expanded in the presence of 50 U/mL recombinant human interleukin-2 (rhIL-2, Novartis, Cambridge, MA, USA) and 0.5 ng/mL interleukin-15 (rhIL-15, Novartis), which was supplemented every 48–72 h during ex vivo expansion for 14–21 days. For antimetabolite studies, 500 nM 6-mercaptopurine (FisherScientific Hampton USA) was administered on day 8 and 10 and 50 nM 6-thioguanine (Sigma, St. Louis, MO, USA) was administered on day 4, 6, 8 and 10.

### 2.6. Knockdown Assay

To analyze siRNA functionality, psiCHECK™-2 vector (Promega, Madison, WI, USA) was used. The psiCHECK™-2 vector consists of both Firefly and Renilla transgene. The Renilla transgene contains a multiple cloning site before the poly(A) sequence allowing the target sequence (CCR5 or Tat/Rev) to be inserted. HEK293 cells (1 × 10^5^) were seeded 16 to 24 h prior to transfection. The CAR containing shRNA (0.05 μg) and psiCHECK™-2 vector with or without target sequence (0.05 μg) were transfected into HEK293T cells using Lipofectamine 3000 (Thermo Fisher, Irwindale, CA, USA). 48 h post-transfection, both Luciferin and Renilla activity were evaluated using a Dual-Luciferase Reporter Kit (Promega). To confirm CCR5 shRNA functionality, U373-MAGIC-CCR5E (0.5 × 10^6^) cells were seeded 16–24 h prior to transfection. The CAR containing CCR5 siRNA (0.05μg) were transfected into cells and analyzed for CCR5 (Biolegend, San Diego, CA, USA) expression by flow cytometry and qualitative RT-PCR.

### 2.7. Protection Assay

CAR-expressing CEM.CCR5^+^ (1 × 10^6^) were infected with either X4- (LAI, NL4-3, YU2) or R5- (Bal) tropical virus and incubated for 18 h before a PBS wash. Cells were resuspended in 1 mL in a 24-well plate (Corning) and cultured for 14 days. On day 7 and 14, supernatant was collected, and HIV expression was determined by an Alliance HIV-1 p24 ELISA (Perkin Elmer, Waltham, MA, USA).

### 2.8. Activation Assay

CAR-expressing Jurkat cells or T cells (1 × 10^5^) were co-cultured with either HEK293 or HEK293.gp160 cells (1 × 10^5^) in a flat-bottom 96-well plate for 24 h. Following incubation, Jurkat, or T cells were collected and analyzed for CD69 or CD137 expression, respectively, by flow cytometry in the MACSQuant Analyzer 10 (Miltenyi, Gaithersburg, ML, USA) and quantified using FlowJo Software (v10). For HIV samples, HIV was neutralized by incubating with a fixation and permeabilization solution for 20 min prior to flow cytometry analysis.

### 2.9. T cell Cytotoxicity Assay

CAR-expressing T cells (1 × 10^5^) were co-cultured with either CEM- or 8E5-expressing GFP cells (4 or 1 × 10^5^) at a 4:1 or 1:1 ratio for healthy or HIV donors, respectively. Following a 4-day incubation, cells were collected and analyzed for tumor survival through GFP expression by flow cytometry in the MACSQuant Analyzer 10 (Miltenyi), and quantified using FlowJo Software (v10). For HIV samples, HIV was neutralized by incubation with the fixation and permeabilization solution for 20 min prior to flow cytometry analysis. Tumor survival was determined by averaging the technical replicates for Mock and setting that value as 100% tumor survival. 

### 2.10. Degranulation Assay

CAR T cells were co-cultured with tumor at a 1:1 ratio in RPMI containing BD GolgiStop^TM^ protein transport inhibitor (BD Bioscience, San Jose, CA, USA) and CD107a antibody (BD Bioscience, San Jose, CA, USA) at 37 °C for 6 h, prior to flow cytometry analysis. LCL-OKT3 was the positive control and KG-1a was the negative control.

### 2.11. Cytokine Production Assay

CAR T cells were co-cultured with tumor at a 1:1 ratio at 37 °C for 4 h before adding Brededlin A GolgiPlug (BD Bioscience, San Jose, CA, USA). Cells were incubated at 37 °C for 24 h before intracellularly stained for INF-γ and TNF-α. LCL-OKT3 was the positive control and KG-1a was the negative control.

### 2.12. Acute Lymphoblastic Leukemia Xenograft Model

All animal experiments were performed under protocols approved by the City of Hope Institutional Animal Care and Use Committee (IACUC). Mice (6–8-week-old NSG mice) were intravenously injected with 5 × 10^5^ Sup-B15 on day 5. Mice were treated with 1 × 10^6^ CAR T cells intravenously. Tumor burden was monitored by live mice imaging using LagoX optical imaging system (Spectral Instruments Imaging, Tucson, AZ, USA). Mice were imaged by injecting XenoLight D-luciferin potassium salt (Perkin Elmer, Waltham, MA, USA) and analyzed on Aura Imaging Software (Spectral Instruments Imaging, Tucson, AZ, USA). Mice were retro-orbitally bled and upon euthanasia, bone marrow and blood were collected for flow cytometry.

### 2.13. Statistics

Analysis was performed using Prism (GraphPad Software Inc., San Diego, CA, USA). The non-parametric Mann–Whitney test, and parametric *t*-test, was applied to group comparisons. The Kaplan–Meier method was applied to survival function estimation and group comparisons. *p*-values of ≤0.05 were considered statistically significant.

## 3. Results

### 3.1. Comparing the Orientation of the Polymerase III Promoter in CAR T cells during shRNA Expression

Although shRNAs have been used in CAR T cells before, we were interested in understanding what optimization must be done to generate an efficacious shRNA-containing CAR. The first optimization that we were interested in was orientation. Previous observations have led to conflicting conclusions as to which orientation, sense or antisense is best for shRNA expression in a CAR construct [19,31]. To test this, we expressed two shRNAs, in either the forward or reverse orientation, under a polymerase III promoter, preceding a second-generation CAR (Figure 1A). We expressed the shRNA under a polymerase III promoter, instead of a standard CAR polymerase II promoter, because it has been shown to have higher expression levels [31]. To compare the orientation, we performed a knockdown assay, in which we expressed Renilla luciferase with either the target-strand (Sense) or the self-strand (Antisense) in the 3′ UTR. As expected, we observed that both shRNAs against the HIV proteins Tat/Rev worked similarly in the forward and reverse orientation in our knockdown assay (Figure 1B).

### 3.2. Wobble Bases Decrease shRNA Self-Targeting

Although orientation does not influence shRNA expression, one of the main concerns when designing shRNA-containing CAR T cells is self-targeting. Self-targeting occurs when short hairpin targets itself, and thereby could decrease the amount of lentivirus that can be produced [32]. To interrogate this, we designed a reverse orientation shRNA against C-C chemokine receptor 5 (CCR5) (Figure 2A), which has previously been implicated in lentivirus suppression [33]. When we tested shCCR5 in our knockdown assay, we observed both the sense and antisense strands were targeted by the shRNA, thus confirming it has self-targeting (Figure 2B). To address this issue, we implemented wobble bases into the shCCR5.

Wobble bases are RNA nucleotides that do not follow the Watson–Crick base pair ruling [34]. For example, guanine can pair with either cytosine or uracil and it has previously been shown that implementing these wobble bases in the shRNA can make the mRNA resistant to shRNA targeting [20]. In the CCR5 shRNA sequence (Figure 2C) we inserted three wobble bases guanine-uracil on the 5′ side near the loop domain to creased wobble base-containing CCR5 shRNA (shCCR5w) (Figure 2D). When we tested the shCCR5w in our knockdown assay, we observed a complete loss of self-targeting without compromising function (Figure 2E).

### 3.3. Optimizing Multi-shRNA Containing CAR

While we have optimized a CAR containing a single shRNA, we were interested if we could put multiple shRNAs into a CAR. Multiple shRNAs would be ideal, as we often want to manipulate multiple aspects of CAR T cells to tailor them to the specific disease. To test this, we designed an anti-HIV CAR with both shCCR5w and shTat/Rev, which would make this CAR HIV-resistant. CCR5 is a known receptor for R5-tropic virus entry [35,36], while Tat and Rev are crucial proteins for HIV replication. Previously we found that it was important to suppress HIV replication because we previously reported that generating CAR T cell products from patients living with HIV (PLWH) has the potential of reactivating HIV during production [37,38]. To generate this HIV-resistant CAR construct, we expressed each shRNA under its own tRNA promoter, in the reverse orientation, preceding the second-generation CAR (Figure 3A). When we checked the shRNA function in our knockdown assay, we observed a similar knockdown effect as we had observed in the single shRNA-containing CAR (Figure 1B, Figure 2E and Figure 3B,C). These data suggest that once the shRNA has been optimized, it is functional regardless of the construct it is put in. Although encouraging that the shRNAs work in a knockdown assay, how these shRNAs function in T cells is still unknown.

To test the shCCR5w CAR, we transduced CCR5-expressing MAGI cells with a conventional CAR or shCCR5w-shTat/Rev CAR. We observed a knockdown of CCR5 expression, although not completely (Appendix A). To confirm, we transduced T cells with conventional CAR and shCCR5w-shTat/Rev CAR and observed knockdown in qPCR and flow cytometry, but similar to the MAGI cells, not completely (Figure 3D,E). To interrogate shTat/Rev’s ability to suppress HIV infection, we transduced CCR5-expressing Jurkat cells with either conventional CAR or shCCR5w-shTat/Rev CAR and then infected them with either an R5-tropic (NL4-3) or X4-tropic (LAI) HIV. Following an 18-h incubation, we collected the supernatant and analyzed HIV production by p24 ELISA. We observed significant suppression of both R5- and X4-tropic virus’ (Figure 3F,G). Although these assays show the function of the shRNAs, how does this impact CAR T cell growth and effectiveness?

We activated and transduced PLWH T cells with either conventional or multiple shRNA-containing N6.CAR. We observed an increased growth of shRNA-containing CAR compared to conventional CAR (Figure 4A), suggesting that HIV reactivation was suppressed leading to less fratricide. To confirm HIV reactivation, we collected supernatant throughout the course of culture and analyzed the p24 levels (Figure 4B). We observed high p24 on day 3, which we attributed to lentivirus addition as no p24 was detected in the mock. We further observed Mock, conventional, and shRNA-containing CAR all had elevated p24 levels on day 9. Interestingly, shRNA-containing CAR have noticeably decreased p24 levels, any more than either mock or conventional CAR. These suggest that shTat/Rev is capable of suppressing HIV reactivation in transduced CAR T cells. Interrogating the phenotype of these cells, we observed comparable CD4/CD8 ratios, as the CD4 numbers could decrease due to N6CAR’s ability to target the gp-160’s expression on HIV-infected cells. Comparing CAR expression in PLWH and healthy donor T cell, on day 7 we observed that multi-shRNA had a comparable CAR expression to conventional CAR in healthy donor, but conventional CAR had a much higher expression in PLWH T cells (Appendix A). However, both multi-shRNA and conventional CAR decreased in PLWH donor at day 20, while the healthy donor stayed consistent (Appendix A). We observed all conditions had a similar exhaustion marker (CD57, PD-1, LAG3, and TIM3) expression, suggesting that the shRNAs did not increase T cell exhaustion. Lastly, we observed similar CD45RA/RO ratios, which suggest that the shRNAs did not influence the T cell phenotype. However, interestingly, we did observe both CARs having higher CD62L levels when compared to the mock, but the shRNA-containing CAR had the highest expression of the memory marker CD62L (Figure 4C). This suggests that adding the CAR to PLWH T cells can retain the naïve and stem cell-like phenotype, more than mock transduced, but the shRNAs enhance retention.

Lastly, we wanted to test if these shRNAs impacted the function of CAR T cells. We transduced healthy donors with either conventional or shCCR5w-shTat/Rev CAR and analyzed activation and killing. To analyze activation, we co-cultured CARs with gp160-expressing HEK cells. Following a 24-h incubation, we observed similar levels of activation between the two CARs (Figure 4D). We observed no activation in the HEK control. To interrogate the killing, we co-cultured CARs with 8E5 cells, which are CEM (T cell lymphoma cell line) that have an endogenous expression of gp120 from a difunctional HIV virus, and cultured them for 4 days [39,40]. Following incubation, we observed similar levels of killing between the two CARs (Figure 4E). These data suggest that shRNA-containing CARs do not have an impact on CAR T cells’ growth or function but can introduce significant resistance to HIV.

### 3.4. Comparing Promoters in the Construction of shRNA-containing CAR

To test the importance of the promoter, we compared a weak tRNA Pol III promoter, which previously has been shown to have comparable efficacy to the strong U6 promoter, but this has only been evaluated in in vitro modeling [41,42]. To compare these promoters, we inserted a short hairpin RNA (shRNA), under either a U6 or tRNALys3 promoter, against the hypoxanthine-guanine phosphoribosyltransferase (HPRT) preceding the second-generation CAR (Figure 5A). We chose HPRT as our target because it has been previously shown that knocking down HPRT constructs cells which are resistant to the antimetabolites 6mercaptopurine and 6thioguanine, and any expression of HPRT leads to a lack of resistance [43,44]. To compare the promoters, we tested them in our knockdown assay. We observed significant knockdown of the target-strands for both promoters (Figure 5B,C). Next, we wanted to test the function of these promoters. To test each promoter’s ability in making cells resistant to antimetabolites, we transfected Jurkat cells with either construct, then administered 1mM 6-thioguanine. We observed only the U6 promoter could resist 6-Thioguanine’s toxicity (Figure 4C,D). Lastly, to confirm that the U6 promoter’s resistance to 6-thioguanine was due to the suppression of HPRT, we performed q-RT-PCR on the resultant Jurkat cells and observed significant suppression of the HPRT RNA (Figure 5F). These data suggest, in the context of generating chemoresistance, that the U6 promoter is not only capable of knocking down HPRT but the knockdown is sufficient for antimetabolite resistance. However, what about its effect on T cells? There has been some evidence that the U6 promoter might be toxic to transduced cells [19].

To interrogate this potential toxicity, we activated and transduced T cells with either conventional or shHPRT containing CD19CAR. We observed that U6-driven HPRT knockdown did not impact CAR T cells growth (Figure 6A). When we interrogated the CAR phenotype at day15, we observed no difference in shHPRT CAR when compared to the conventional CAR (Figure 6B). Next, we wanted to see if the knockdown of HPRT impacted the functionality of CD19CAR T cells, in vitro. We co-cultured conventional or shHPRT CD19CAR T cells with CD19-expressing tumors and analyzed activation, cytokine production, degranulation, or tumor killing. When we analyzed CAR T cell activation, we observed comparable levels of CD69, CD137, and CD25 expression of shHPRT CAR T cells when compared to conventional CAR (Figure 6C and Appendix A). The shHPRT CAR had comparable cytokine production for both interferon-gamma and tumor necrosis factor-alpha when compared to conventional CAR (Appendix A). For tumor killing, we observed shHPRT CAR T cells had equal levels of degranulation, as observed through CD107a expression (Appendix A), and the elimination of tumor in killing assay (Figure 6D). All of this suggests that the U6 promoter-induced knockdown in CAR T cells does not impact the growth or functionality of the CAR T cells, but how does it influence CAR T cells in vivo?

To compare shHPRT CAR T cells to conventional CAR, in vivo, we used our acute lymphoblastic leukemia (ALL) xenograft mouse model [4]. In NSG mice, we engrafted the ALL cell line, SupB15, and allowed 5 days of incubation before we treated the mice with CAR T cells. We observed both the shHPRT and conventional CAR have similar efficacy as determined by live mice imaging (Figure 6E,F). These data suggest that the knocking down of HPRT in T cells did not impact trafficking or function in vivo. To analyze persistence, we collected blood and bone marrow upon euthanasia (~day 75) and analyzed for CAR T cells. We observed a higher, although not significant, percentage of CAR T cells in the shHPRT group when compared to the conventional CAR (Figure 6G). These data suggest that knocking down HPRT in CAR T cells does not impact CAR T cells’ functionality or efficacy.

To test T cell function following antimetabolite treatment, we activated and transduced T cells with shHPRT CAR and added two doses of 6-mercaptopurine (6MP) (500 nM) at days 8 and 10 (Figure 7A). 6-mercaptopurine is an antimetabolie that is processed by HPRT into thio-guanine triphosphate, which will be incorporated into the cell’s DNA, leading to apoptosis [45]. We found two treatments of 500nM 6MP had the best enrichment of shHPRT CAR (Figure 7B–D). We observed no change in the phenotype of shHPRT CAR T cells following either 6MP treatment (Figure 7E and Appendix A). Next, we wanted to see if the enrichment impacted shHPRT CAR T cell function. Co-culturing shHPRT CAR T cells with and without 6MP treatment with target cells, we observed 6MP-treated CAR T cells had similar activation (CD69 and CD25) (Figure 7F and Appendix A), cytokines production (INF g and TNFα) (Appendix A), and degranulation (CD107a) (Figure 7G) to non-treated shHPRT CAR T cells. Translating in vivo, we observed both non-- and 6MP-treated shHPRT CAR T cells had similar efficacy as determined by imaging live mice (Figure 7H,I). To analyze persistence, we collected blood and bone marrow upon euthanasia (~day 70) and looked for CAR T cells. We observed a higher, although not significant, percentage of CAR T cells in the non-treatment group when compared to the 6MP-treated shHPRT CAR (Figure 7J). We also confirm this was not 6MP or CD19.CAR specific by treating shHPRT anti-HIV CAR T cells with 6TG (Appendix A). We observed, like 6MP, that 6TG-treated CAR T cells had no impact in phenotype (Appendix A), activation (Appendix A), and target killing (Appendix A). These data suggest that the shRNA-driven knockdown by the U6 promoter in CAR T cells is not toxic to the cells, and HPRT knockdown can make CAR T cells resistant to antimetabolite treatment, thus allowing for potential ex vivo enrichment or in vivo chemotherapy.

## 4. Discussion

Expanding CAR T cell therapy to diseases beyond hematological malignancies faces significant obstacles, including harsh conditions such as tumor microenvironments, exhaustion, viral infection, and fratricide. To overcome these challenges, genetically altering the CAR T cell with RNA-guided nucleases such has CRISPR-Cas9, has been a promising strategy, preclinically [46,47]. Furthermore, CRISPR has been used preclinically to knockout exhaustion makers, such as PD-1, and proteins needed for HIV infection, such as CCR5, which has alleviated these respective conditions [48]. However, nuclease gene editing use methods, such as electroporation or nanoparticles, modify the established CAR T cells manufacturing processes. Although it appears that these methods do not impact CAR T cell fitness, this area remains largely unstudied [12,13]. In this study, we outline an alternative strategy that genetically targets CAR T cells specifically without altering the current manufacturing platform.

Short interfering RNAs (siRNAs) offer similar capabilities to RNA-guided nucleases in knocking down genes that could be detrimental to CAR T cells efficacy [17]. However, unlike other CRISPR, shRNAs also offer the potential to enhance CAR T cells specifically. For example, Zhang et al. recently published using a shRNA to target lysine-specific demethylase 1 (LSD1), which plays a role in T cell infiltration [16,49]. Kang et al. demonstrated that a shRNA against IL-6 could dampen cytokine release syndrome-like symptoms [21]. Additionally, shRNA technology has been used to create allogenic CAR T cells, which are currently being tested clinically (NCT04613557). We contribute to this body of work by showing two mechanisms in which shRNAs within the CAR construct can offer unique potentials for CAR T cell therapies. Firstly, we show that an shRNA against HIV replication proteins Tat and Rev will prevent both R5- and X4- tropic virus replication and will also suppress some HIV reactivation that could not be addressed with knocking out CCR5 alone and is unobtainable without persistent shRNA expression (Figure 4). Secondly, we show that we can make CAR T cells specifically resistant to antimetabolites, 6MP and 6TG, which allowed for ex vivo enrichment that is not possible by current manufacturing protocols (Figure 5 and Figure 6). Additionally, shRNA against HPRT shows the potential for developing chemo-resistant CAR T cells that could be used in combination therapy. Although shRNAs offer many potentials for CAR therapy, they require far more optimization than nucleases.

In this this study, we described four aspects/questions regarding orientation, self-targeting, multiple shRNAs, and promoter strength that need to be addressed when developing a shRNA-containing CAR construct. Of these, self-targeting poses the highest potential problem as it would decrease lentivirus production and could lead to decreased functionality once transduced [20]. Although we did observe some self-targeting in all our shRNAs, only shCCR5 posed a potential problem (Figure 5). We were easily able to alleviate this issue by using wobble bases in the sense arm of the shRNA, which is a well-known way of alleviating shRNA targeting [20]. Aside from self-targeting, none of the other parameters influenced the function of the shRNA. Interestingly, An et al. determined that the U6 promoter, when compared to the H1 promoter, was more efficacious in knocking down CCR5 in human lymphocytes, which could be the reason we see some suppression but not complete knockdown in T cells (Figure 2) [19]. Additionally, Lee et al. also demonstrates that a recombination within the construct does not occur to an appreciable extent [22]. Furthermore, although shRNA expression off a pol II promoter has been tested, it largely avoids the CAR T cell space due to potential self-targeting [50,51]. This is impactful because although other labs have utilized shRNAs in their CAR constructs, no one has dissected the optimization of the shRNAs used in their strategies [18,21,22]. Overall, incorporating shRNAs in the CAR construct presents a promising alternative to nuclease driven knockouts, as it does not disrupt established manufacturing platforms and offers the potential to target certain aspects of non-hematological malignancies that nucleases cannot address.

## 5. Conclusions

This article shows an alternative method for editing CAR T cells by introducing shRNAs into the CAR construct. Although this method has been used before, this article addresses the optimizations necessary to have functional shRNA-containing CAR T cells. Once optimized, the shRNAs offer CAR specific knock downs that can enhance CAR functionality. Furthermore, shRNA offers knock downs that could protect CAR T cells from harmful agents, such as HIV and chemotherapy. In conclusion, this article provides guidance to adding optimized shRNAs to CAR T cells.

## 6. Patents

R.U., T.S., and K.V.M. hold a patent on shRNAs for anti-HIV CAR T cells and X.W. and S.J.F. holds a patent on N6CAR. 

## Figures and Tables

**Figure 1 cancers-15-02848-f001:**
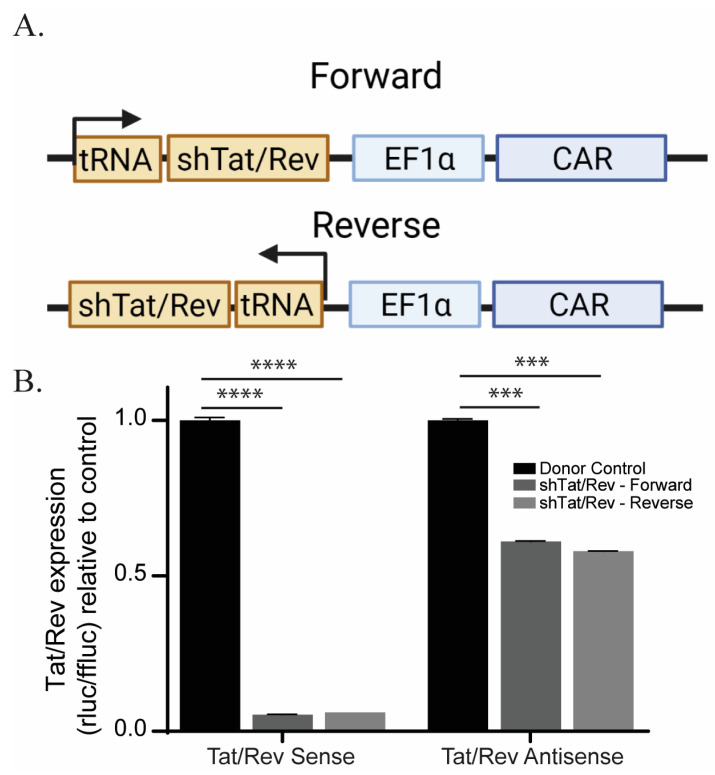
shRNA directionality in the CAR construct does not impact shRNA function. (**A**) Schematic of CAR containing shTat/Rev in forward and reverse orientation. (**B**) Knockdown analysis (*n* = 4) of target (Sense) and self (Antisense) strands of shTat/Rev as determined by psicheck plasmid. To ensure accuracy and reliability, each experiment had technical replicates (*n*). This figure represents 1 biological replicate of the three that were performed. The symbols **** and *** represent a *p* value of <0.0001 and 0.0005 respectively.

**Figure 2 cancers-15-02848-f002:**
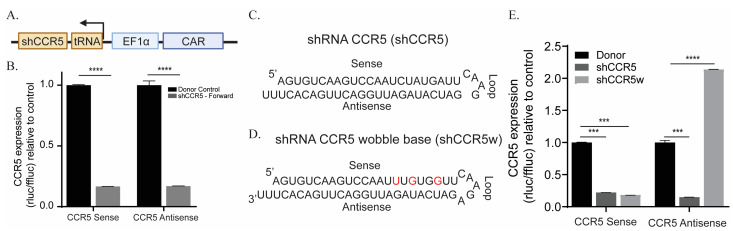
Wobble bases alleviate shRNA self-targeting. (**A**) Schematic of CAR containing shCCR5 in reverse orientation. (**B**) Knockdown analysis (*n* = 4) of target (sense) and self (antisense) strands of shCCR5 as determined by psicheck plasmid. Schema of the original shCCR5 sequence (**C**) and wobble base containing shCCR5 (**D**). (**E**) Confirmation through knockdown assay of diminished of self-targeting by shCCR5w, modified bases are in red. To ensure accuracy and reliability, each experiment had technical replicates (*n*). This figure represents one biological replicate of the three that were performed. The symbols **** and *** represent a *p* value of <0.0001 and 0.0005 respectively.

**Figure 3 cancers-15-02848-f003:**
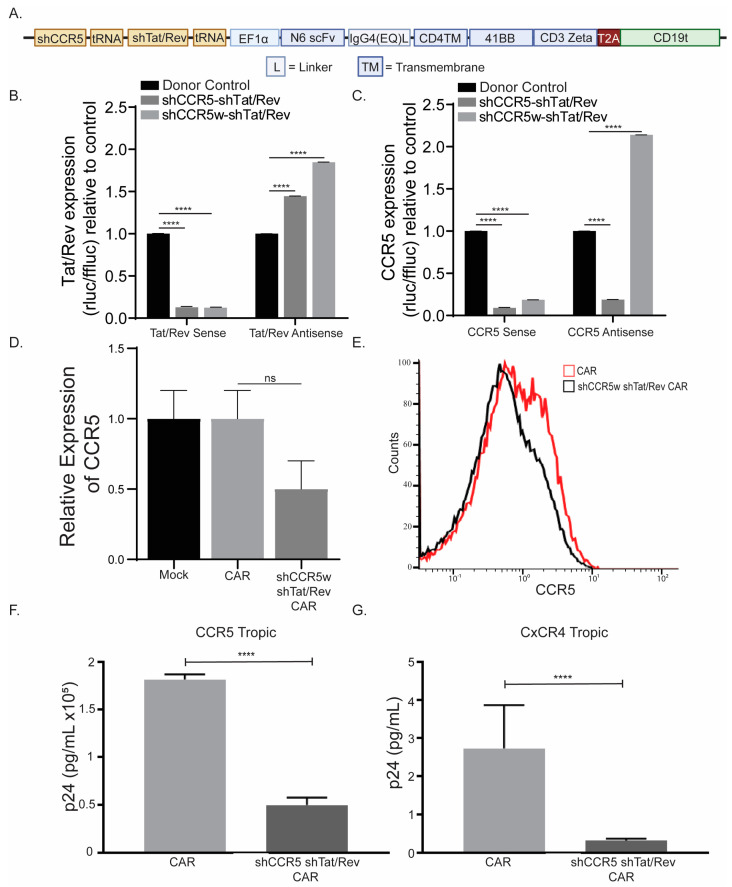
Multiple shRNAs in CAR construct do not impact shRNA function. (**A**) Schematic of shCCR5w and shTat/Rev CAR construct. Knockdown analysis of target (sense) and self (antisense) strands of shTat/Rev (*n* = 4) (**B**) and shCCR5 (*n* = 4) (**C**) as determined by psicheck plasmid. CCR5 knockdown efficiency in healthy donor T cells as determined by qPCR (*n* = 3) (**D**) and flow cytometry (*n* = 1) (**E**). Inhibition of HIV replication of R5- (**F**) and X4-tropic (**G**) virus as determined by p24 analysis. To ensure accuracy and reliability, each experiment had technical replicates (*n*). This figure represents one healthy donor of the three healthy donors that were interrogated. The symbols **** and ns represent a *p* value of <0.0001 and non-significance, respectively.

**Figure 4 cancers-15-02848-f004:**
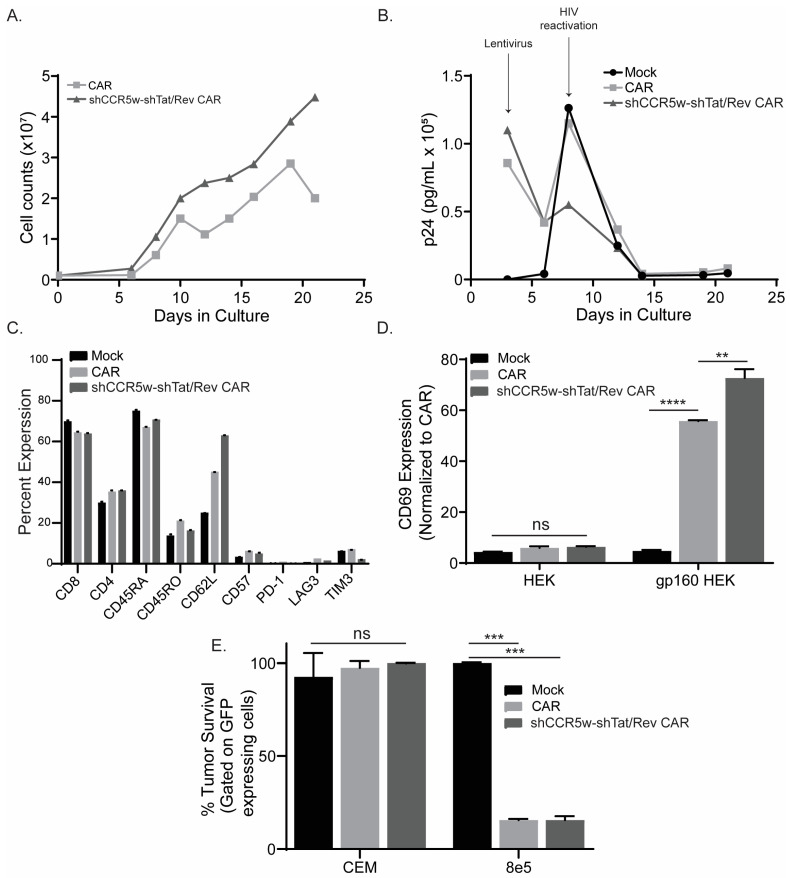
Multiple shRNAs in CAR construct do not impact CAR functionality. (**A**) Growth curve comparing shCCR5w-shTat/Rev anti-HIV CAR against conventional anti-HIV CAR in PLWH T cells. (**B**) p24 analysis of HIV reactivation in PLWH T cells through the course of culture comparing shCCR5w-shTat/Rev anti-HIV CAR, conventional anti-HIV CAR, and mock. (**C**) Phenotype analysis of shCCR5w-shTat/Rev anti-HIV CAR, conventional anti-HIV CAR, and mock for PLWH T cells using flow cytometry. Functional analysis, CD69 activation (**D**), and killing (**E**) of shCCR5w-shTat/Rev anti-HIV CAR, conventional anti-HIV CAR, and mock (*n* = 3). To ensure accuracy and reliability, each experiment had technical replicates (*n*). This figure represents one healthy or PLWH donors of the three healthy or PLWH donors that were interrogated. The symbols ****, ***, **, and ns represent a *p* value of <0.0001, 0.0005, 0.001, and non-significance, respectively.

**Figure 5 cancers-15-02848-f005:**
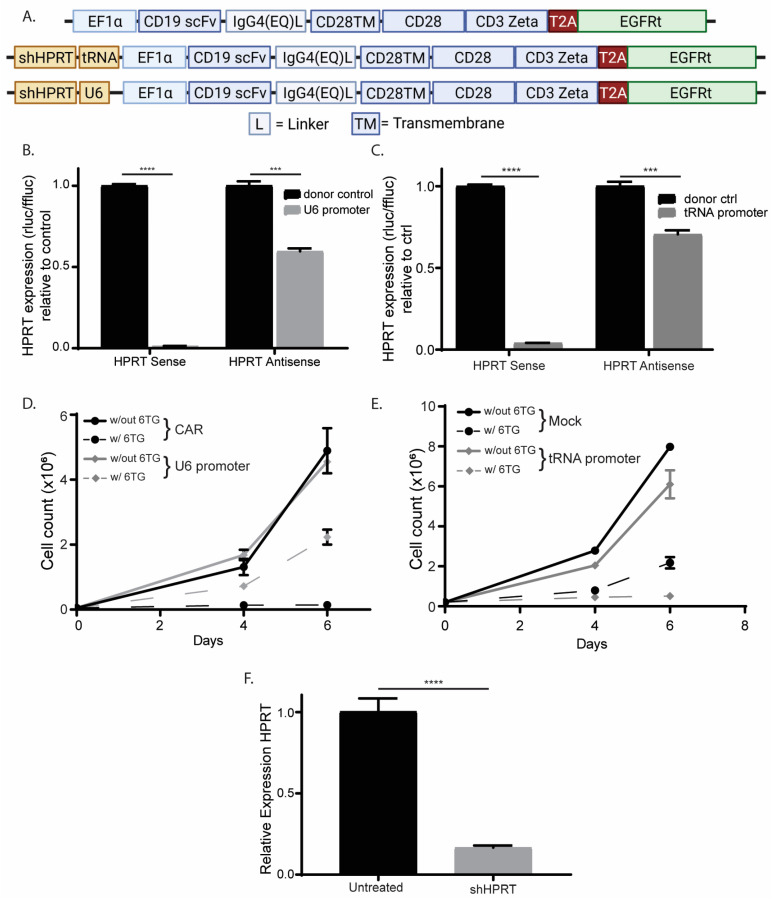
Strong U6 promoter is necessary for the development of chemo-resistant CAR. (**A**) Schematic of shHPRT CD19CAR with either weak tRNA or strong U6 promoter. Knockdown analysis of target (sense) and self (antisense) strands of U6 promoter (*n* = 4) (**B**) or tRNA promoter (*n* = 4) (**C**) as determined by psicheck plasmid. Resistance of shHPRT to 6thioguanine in Jurkat cells for U6 promoter (*n* = 2) (**D**) or tRNA promoter (*n* = 2) (**E**). (**F**) qPCR analysis of HPRT expression in Jurkat cells for U6 promoter (*n* = 3). To ensure accuracy and reliability, each experiment had technical replicates (*n*). This figure represents one biological replicate of the three that were performed. The symbols **** and *** represent a *p* value of <0.0001 and 0.0005 respectively.

**Figure 6 cancers-15-02848-f006:**
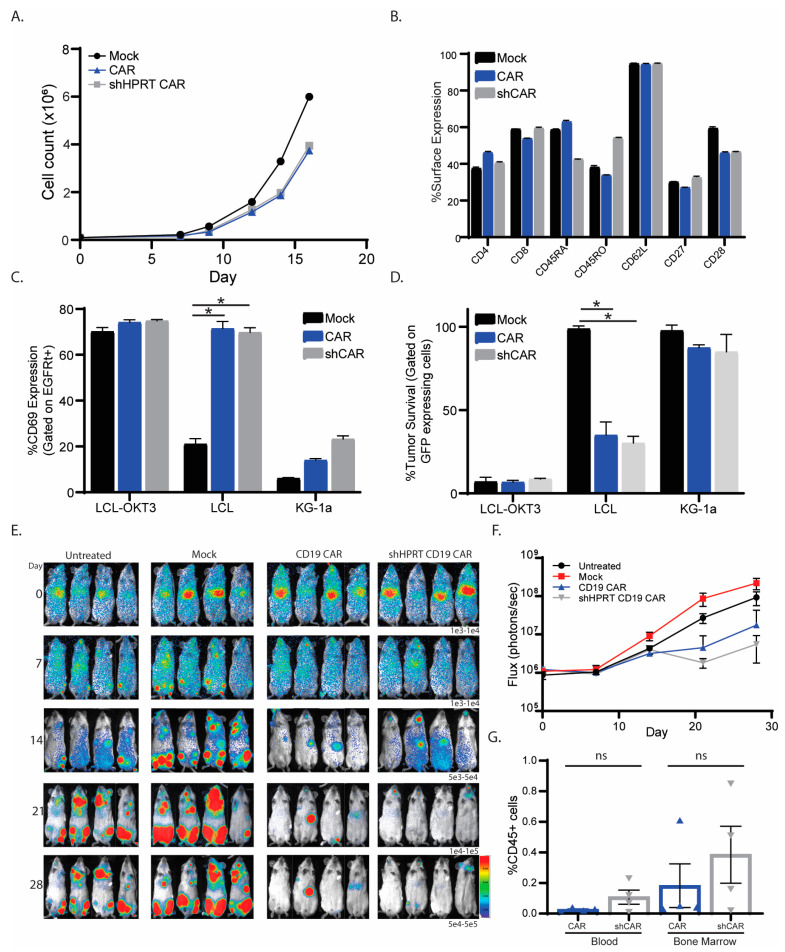
U6 promoter does not impact CAR T cell growth or function. (**A**) Growth curve of shHPRT CAR when compared to mock and conventional CAR. (**B**) Phenotype of shHPRT CAR compared to mock and conventional CAR. Functional analysis, CD69 activation (**C**) and killing (**D**) of shHPRT CAR, conventional CAR, and mock (*n* = 2). E. Mice were injected with 0.5 × 10^6^ SupB15 cells 5 days prior to CAR treatment. On day 5 mice were injected with 1 × 10^6^ shHPRT CAR or conventional CAR (*n* = 4). Tumor growth was analyzed by live cell imaging (**E**) and graphed based on flux (**F**). (**G**) Upon euthanasia, blood and bone marrow were collected and analyzed for persistence. Mock and untreated were used as control. To ensure accuracy and reliability, each experiment had technical replicates (*n*). This figure represents one healthy donor of the seven healthy donors that were interrogated. The symbols * and ns represent a *p* value of 0.05 and non-significance respectively.

**Figure 7 cancers-15-02848-f007:**
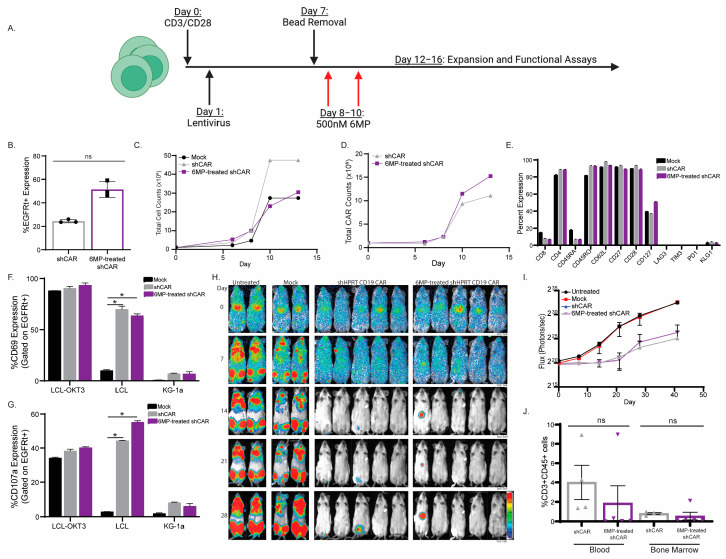
6mercaptopurine-enriched shHPRT CAR T cells have comparable function to unenriched shHRPT CAR T cells. (**A**) Schema of 6-mercaptopurine enrichment of shHPRT CAR T cells. (**B**) 6mercaptopurine enrichment of shHPRT CAR. Growth curve of 6-mercaptopurine treated shHPRT CAR T cells as compared to mock and untreated shHPRT CAR for total T cells (**C**) and total CAR (**D**). (**E**) Phenotype analysis of 6mercaptopurine treated shHPRT CAR T cells when compared to untreated. Functionality analysis, CD69 activation (**F**), and CD107a degranulation (**G**), of 6-mercaptopurine, treated and untreated shHPRT CAR T cells (*n* = 3). Mice were injected with 0.5 × 10^6^ SupB15 cells 5 days prior to CAR treatment. On day 5, mice were injected with 1 × 10^6^ shHPRT CAR or conventional CAR (*n* = 5). Tumor growth was analyzed by live cell imaging (**H**) and graphed based on flux (**I**). (**J**) Upon euthanasia, blood and bone marrow were collected and analyzed for persistence. Mock and untreated were used as control. To ensure accuracy and reliability, each experiment had technical replicates (*n*). This figure represents one healthy donor of the three healthy donors that were interrogated. The symbols * and ns represent a *p* value of 0.05 and non-significance respectively.

**Table 1 cancers-15-02848-t001:** Short hairpin RNA sequences.

Short Hairpin RNA	Sequence
shCCR5	AGTGTCAAGTCCAATCTATGATTCAAGAGATCATAGATTGGACTTGACACTTTTTTT
shCCR5w	AAAAAAAGTGTCAAGTCCAATCTATGATCTCTTGAACCACAAATTGGACTTGACACT
shTat/Rev	GCGGAGACAGCGACGAAGAGCTTTGTGTAGGCTCTTCGTCGCTGTCTCCGCTTTTTT
shHPRT	GCACTGAATAGAAATGGTGGTTCAAGAGATCACTATTTCTATTCAGTGCTTTTT

## Data Availability

All the data associated with the study is in the paper or is provided in the Appendix A.

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
