# Peer review of "Evaluation of the Elements of Short Hairpin RNAs in Developing shRNA-Containing CAR T Cells"

_cancers, 2023, doi:10.3390/cancers15102848_

Round 1

Reviewer 1 Report

This is a lovely paper; clear, concise, easy to read, great figures with a significant conclusion. I enjoyed reading the paper!

BUT, here is a major question in my mind regarding the replication of experiments. Many of results have essentially no range while in some cases the Mock experiments where there is presumably no interventions, there are significant ranges in values. This does not make sense to me. 

There is replication of whole experiments - that is what we want to see because it reflects on the reproducibility of data and conclusions. The alternative, replication of assays only testifies to the competence of the person making the measurements.

Detailed Suggestions and comments

29: should words in the title be repeated in the Key Words?

45: which may cause

46: cells, which

57:  incorporated into a CAR construct to lower expression of specific genes to optimize activity of transduced cells (15-17)  (This combines two sentences with the same idea)

73: constructs …. were generated..

96: sentence unclear – maybe “stained with intracellular antibodies to IFg and TNFa…..

115: hygromycin

116: (brain lymphoma)???

119: 10E9 cells/

124: [Note - In centrifugation it’s the g-value that counts; giving the rpm without the centrifuge type does not define the spin-down conditions]

134: the HEK cells seem to be designated as either HEK293 (earlier) or 293T (here) suggesting cells used were HEK293T cells; regardless, check cell designations.

161: Omit “was determined”

177: What are 8e5 GFP-expressing cells

183: Degranulation Assay(?)

205: method was

216: Sentence is confusing because it’s not clear whether the clause “which is widely used….” refers to the Pol III promoter or some other Pol II promoter. There are no “standard” Pol II promoters - high activity Pol II promoters (like CMV and EF1a) are used and regulatable Pol II promoters of many types are used.  Pol II promoters transcribe long sequences whereas Pol III transcribe relatively short sequences, which is important regarding the cell’s response to the different types of RNA.

216, 223, and throughout the paper: In the text panels are in lower case whereas in the figures they are capitalized.

221: You could add the words “As expected, “ because the observations are just that.

Figs. 1, 2, 3, etc.: Really nice data. The error bars look “too good to be true” if they reflect independent experiments rather than multiple measurements from the same experiment. It’s rare that the same levels of transfection/transduction are so close.  Multiple assays from the same experiment do not give a good idea of the range of KD that might occur.

230: Self-targeting occurs when the short hairpin targets itself and thereby

244: You could be more specific and say in RNA,  G can pair with either C or U (key to the “Universal Genetic Code”), which would explain the logic of your highlighted G’s and U. (It’s nice to see references that are seminal, pre 2000)

264 etc.: These data (suggest rather than suggests)

Fig3: Need to define the cells in the legend. It’s a mystery why the relative expression of CCR5 varies so much (Panerl E) when in other assays there is essentially no variation.  What is the definition of “Relative” if not to either Mock or CAR cells without sh constructs. The Y axis of Panel E needs better definition.

276: transduced

283: replace function with effectiveness, which then gives meaning to the second use of the word.

302: Rewrite the sentence because it suggests CARs are cells with phenotype

Fig. 4E: Mock should be at 100 (and since untouched, should have the lowest range rather than the highest range) if the other results are normalized to it.

320: It would be helpful to clarify the use and definition of  CEM and 8e5 cells

321: but can introduce significant resistance to HIV.

324: to test the importance of promoter

325: Sentence says both weak and strong Pol III promoters have equal efficacy. This suggests promoter strength is not important; unless efficacy is referring to ???

Fig. 6B & C, define % expression for y-axis

387: Define 6MP in the text (rather than figure legend) and why it was selected

396: imaging live mice

420: Supp Fig. 1??

444: …with RNA-guided nucleases such as CRISPR-Cas9 has…

453: …to RNA-guided nucleases

471: …four aspects/questions regarding  (…

466: Only 1 anti-metabolite (6MP) was tested; resistant to an antimetabolite like 6MP..

479: U6 promoter compared to itself??

484: Also demonstrates that recombination within the construct does not occur to an appreciable extent.

Author Response

Reviewer 1

This is a lovely paper; clear, concise, easy to read, great figures with a significant conclusion. I enjoyed reading the paper!

BUT, here is a major question in my mind regarding the replication of experiments. Many of results have essentially no range while in some cases the Mock experiments where there is presumably no interventions, there are significant ranges in values. This does not make sense to me. 

There is replication of whole experiments - that is what we want to see because it reflects on the reproducibility of data and conclusions. The alternative, replication of assays only testifies to the competence of the person making the measurements.

Thank you for your positive review of our manuscript! We would like to clarify that all experiments were repeated three times with technical replicates. However, we understand that the representative experiments we displayed may have seemed too good to be true. To account for donor differences, we made the decision not to combine all experiments together. To alleviate concerns we have added information about the number of independent experiments to all figure legends.  We hope that this clarification has addressed all of the concerns raised in the manuscript.  Changes have been highlighted in the manuscript.

Detailed Suggestions and comments

29: should words in the title be repeated in the Key Words?

45: which may cause

46: cells, which

57:  incorporated into a CAR construct to lower expression of specific genes to optimize activity of transduced cells (15-17)  (This combines two sentences with the same idea)

73: constructs …. were generated..

96: sentence unclear – maybe “stained with intracellular antibodies to IFg and TNFa…..

115: hygromycin

116: (brain lymphoma)???

119: 10E9 cells/

124: [Note - In centrifugation it’s the g-value that counts; giving the rpm without the centrifuge type does not define the spin-down conditions]

134: the HEK cells seem to be designated as either HEK293 (earlier) or 293T (here) suggesting cells used were HEK293T cells; regardless, check cell designations.

161: Omit “was determined”

177: What are 8e5 GFP-expressing cells

183: Degranulation Assay(?)

205: method was

216: Sentence is confusing because it’s not clear whether the clause “which is widely used….” refers to the Pol III promoter or some other Pol II promoter. There are no “standard” Pol II promoters - high activity Pol II promoters (like CMV and EF1a) are used and regulatable Pol II promoters of many types are used.  Pol II promoters transcribe long sequences whereas Pol III transcribe relatively short sequences, which is important regarding the cell’s response to the different types of RNA.

216, 223, and throughout the paper: In the text panels are in lower case whereas in the figures they are capitalized.

221: You could add the words “As expected, “ because the observations are just that.

Figs. 1, 2, 3, etc.: Really nice data. The error bars look “too good to be true” if they reflect independent experiments rather than multiple measurements from the same experiment. It’s rare that the same levels of transfection/transduction are so close.  Multiple assays from the same experiment do not give a good idea of the range of KD that might occur.

All experiments were repeated three times. The figures shown are representative of technical replicates. The n in the figure legends represents technical replicates while in figure legends we wrote

230: Self-targeting occurs when the short hairpin targets itself and thereby

244: You could be more specific and say in RNA,  G can pair with either C or U (key to the “Universal Genetic Code”), which would explain the logic of your highlighted G’s and U. (It’s nice to see references that are seminal, pre 2000)

264 etc.: These data (suggest rather than suggests)

Fig3: Need to define the cells in the legend. It’s a mystery why the relative expression of CCR5 varies so much (Panerl E) when in other assays there is essentially no variation.  What is the definition of “Relative” if not to either Mock or CAR cells without sh constructs. The Y axis of Panel E needs better definition.

We have updated the Y-axis in panel E. It is a histogram so the why axis is just showing the height while the X-axis is the expression of CCR5. There is no mock in this figure it is only comparing CAR T cells and shRNA containing CAR.

276: transduced

283: replace function with effectiveness, which then gives meaning to the second use of the word.

302: Rewrite the sentence because it suggests CARs are cells with phenotype

Fig. 4E: Mock should be at 100 (and since untouched, should have the lowest range rather than the highest range) if the other results are normalized to it.

Since we performed technical replicates in this experiment, we normalized the data by averaging the tumor survival of the mock and setting that to 100%. Because we do this mock has the potential of having some fluctuation due to variability in each well.

320: It would be helpful to clarify the use and definition of CEM and 8e5 cells

321: but can introduce significant resistance to HIV.

324: to test the importance of promoter

325: Sentence says both weak and strong Pol III promoters have equal efficacy. This suggests promoter strength is not important; unless efficacy is referring to ???

Fig. 6B & C, define % expression for y-axis

387: Define 6MP in the text (rather than figure legend) and why it was selected

396: imaging live mice

420: Supp Fig. 1??

444: …with RNA-guided nucleases such as CRISPR-Cas9 has…

453: …to RNA-guided nucleases

471: …four aspects/questions regarding  (…

466: Only 1 anti-metabolite (6MP) was tested; resistant to an antimetabolite like 6MP..

479: U6 promoter compared to itself??

484: Also demonstrates that recombination within the construct does not occur to an appreciable extent.

Reviewer 2 Report

The submitted review article “Evaluation of the elements of short haripins RNAs in developing shRNA-containing CAR T cells” by Urak et al. reports on optimization of several elements of shRNAs to better knock down expression of genes in CAR T cells. Overall, the manuscript is well written and represents an important step in advancing the gene knockdown approach to modulating CAR T cell function. The only major criticism of the manuscript I have is that it under-reports details of data shown in figures (i.e. number of biological replicates, do graphs show representative data or an average of several biological replicates?, statistical tests used and what stars mean, etc.) and that many pieces of data lack statistical testing. These issues would need to be addressed to evaluate claims in the text regarding comparison of conditions. A few additional points below would improve the manuscript, but are relatively minor:

1.    For each condition tested, it would be good to show expression of the CAR on the cell surface so the reader could determine if changes in how the shRNA is expressed alters CAR transcription.

2.    Additional detail regarding flow cytometry analysis of GFP expression in the T cell cytotoxicity assay section of the methods (page 5, lines 176-182) would clarify what the assay is actually measuring since a 4-day cytotoxicity assay is not typical. Is GFP expression a precent of all cells in the well (CAR T cells + tumor cells)? If so how, are potential differences in CAR T cell expansion between conditions accounted for?

3.    Define cell types used – what are CEM & 8e5 cells?  Why were they chosen? What is a PLWH T cell?

4.    An experiment (in vitro or in vivo) combining shHPRT CAR T cells with an antimetabolite would help drive home the potential utility of shRNA containing CAR T cells.

5.    Additional discussion of RNA pol II vs. RNA pol III promoters use here in driving shRNA expression is warranted.  Theoretically, the CAR and shRNA could come from a single Pol II driven transcript… are there advantages/disadvantages to this vs. inclusion of additional pol III promoters?
